# *RPE65* c.353G>A, p.(Arg118Lys): A Novel Point Mutation Associated with Retinitis Pigmentosa and Macular Atrophy

**DOI:** 10.3390/ijms231810513

**Published:** 2022-09-10

**Authors:** Mirjana Bjeloš, Mladen Bušić, Ana Ćurić, Borna Šarić, Damir Bosnar, Leon Marković, Biljana Kuzmanović Elabjer, Benedict Rak

**Affiliations:** 1Department of Ophthalmology, Reference Center of the Ministry of Health of the Republic of Croatia for Pediatric Ophthalmology and Strabismus, University Hospital “Sveti Duh”, 10000 Zagreb, Croatia; 2Faculty of Medicine, Josip Juraj Strossmayer University of Osijek, 31000 Osijek, Croatia; 3Faculty of Dental Medicine and Health Osijek, Josip Juraj Strossmayer University of Osijek, 31000 Osijek, Croatia

**Keywords:** retinal dystrophies, night blindness, retinitis pigmentosa, genetic therapy

## Abstract

Precise genetic diagnosis in *RPE65*-mediated retinitis pigmentosa (RP) is necessary to establish eligibility for genetic treatment with voretigene neparvovec: a recombinant adeno-associated viral vector providing a functional *RPE65* gene. This case report aims to report a novel RP-related point mutation *RPE65* c.353G>A, p.(Arg118Lys), a variant of uncertain significance associated with a severe clinical presentation and the striking phenotypic feature of complete macular atrophy. We report the case of a 40-year-old male with inherited retinal dystrophy, all features typical for the *RPE65*-associated RP, and marked macular atrophy. Genetic testing identified that the patient was a compound heterozygote in *trans* form with two heterozygous variants: *RPE65* c.499G>T, p.(Asp167Tyr) and *RPE65* c.353G>A, p.(Arg118Lys). Furthermore, short-wavelength and near-infrared autofluorescence patterns exhibited deficiencies specific to mutations in the visual cycle genes. To the best of our knowledge, *RPE65* c.353G>A, p.(Arg118Lys) is the first described point mutation on this locus, among all other reported insertional mutations, currently classified as likely benign and of uncertain significance. We concluded that this variant contributed to the pathological phenotype, demonstrating its significance clearly to be reclassified as likely pathogenic. This being the case, patients with this specific variant in homozygous or compound heterozygous form would be likely candidates for genetic treatment with voretigene neparvovec.

## 1. Introduction

The *RPE65* gene (retinoid isomerohydrolase OMIM ID 180069) encodes a protein that is the source of isomerohydrolase activity and catalyzes the conversion of the all-*trans* retinyl ester to 11-*cis* retinol in the retinal pigment epithelium (RPE) [1,2]. *RPE65* pathogenic variants are associated with autosomal recessive retinitis pigmentosa (RP) 20, Leber congenital amaurosis (LCA) 2, and autosomal dominant RP 87 with choroidal involvement [2].

*RPE65*-mediated RP is a sight-threatening genetic disorder causing a severe form of rod-cone inherited retinal dystrophy (IRD) that eventually progresses to complete blindness [3,4,5].

In this clinically and genetically heterogeneous group of IRDs, precise genetic diagnosis is required to establish eligibility for genetic treatment of *RPE65*-associated IRD with voretigene neparvovec: a recombinant adeno-associated viral vector providing a functional *RPE65* gene to act in place of a mutated *RPE65* gene [6].

This case report aims to report a novel RP-related point mutation *RPE65* c.353G>A p.(Arg118Lys), a variant of uncertain significance (VUS) associated with a severe clinical presentation and the striking phenotypic feature of complete macular atrophy. This being the case, patients with this specific variant in homozygous or compound heterozygous form are likely candidates for genetic treatment with voretigene neparvovec.

## 2. Case Presentation

A 40-year-old male with IRD was referred to our clinic for clinical examination and genetic testing. Since the age of three, his mother noticed that he could not see well in the dark. The patient himself noticed a decline in vision throughout his life, but at the age of 30, a more pronounced deterioration ensued, progressively worsening over the following years. His family history was unremarkable.

On clinical examination, his best corrected visual acuity (BCVA) was 1.1 logMAR binocularly at distance (tested at 1 m) and 1.0 logMAR at near (tested at 40 cm). The right eye (RE) measured 1.2 logMAR at 1 m and 1.0 logMAR at 40 cm, while the left eye (LE) demonstrated 1.6 logMAR distance (tested at 0.5 m) and 1.5 logMAR near (tested at 10 cm). He could not perform the CSV-1000 contrast sensitivity test. Standardized color vision tests (Farnsworth’s D-15 dichotomous test and Lanthony desaturated 15-hue panel) showed multiple errors without single-axis dominance. Goldmann kinetic perimetry using III4 revealed residual islands of vision within the central 10° with a total of 232° and 196° at RE and LE, respectively. Octopus^®^ (Haag-Streit Inc., Mason, OH, USA) static G1 perimetry evidenced markedly reduced retinal sensitivity within the central 30°: mean sensitivity measured RE 0.3 dB and LE 0.2 dB. MAIA microperimetry (iCare Finland Oy, Vantaa, Finland), a scotoma finder strategy, presented a scotoma area within the central 20° with only a few single loci of markedly reduced retinal sensitivity within the temporal and inferior margins of chorioretinal atrophy in both eyes (BE). Fixation was absolutely unstable, measuring P1 25% and P2 66% on RE, and relatively unstable, measuring P1 32% and P2 78% on LE. HRA+ OCT Spectralis^®^ (Heidelberg Engineering, Heidelberg, Germany) depicted areas of retinal atrophy in the central 20° measuring 14.75 and 14.45 mm^2^, with a central macular thickness of 222 and 224 µm on RE and LE, respectively, with a clear view of the choroidal vessels on BE (Figure 1).

The short-wavelength fundus autofluorescence (SW-AF) was absent, as was near-infrared reflectance (NIR), indicating visual cycle (VC) gene mutations [7] (Figure 2). SW-AF is a marker for retinal pigment epithelium (RPE) lipofuscin, while NIR is used for the detection of melanin in RPE and choroid. Due to the loss of RPE and choroid, with the exception of very large choroidal vessels, SW-AF and NIR share the same characteristics: increased tissue transparency so that the scleral reflectance is clearly visualized (Figure 2).

Optos^®^ California (Optos Inc., Marlborough, MA, USA) ultra-widefield imaging depicted a grayish optic nerve head with clear boundaries. Chorioretinal atrophy was present on BE in the macular area, measuring 4.3 × 4.2 mm in RE and 4.4 × 4.0 mm in LE. In the mid and far periphery, seals of chorioretinal atrophy were evident, with marginal partial pigmentation ranging from 0.5 to 2-3 DD RE to 7-8 DD LE. Blood vessels were preserved up to the midperiphery, further markedly thinned (Figure 3).

Full-field electroretinography (FFERG) testing (Roland Consult RETI-port/scan 21, Roland Consult Stasche and Finger GmbH–German Engineering, Brandenburg an der Havel, Germany) according to ISCEV standards depicted extinguished scotopic and photopic responses on BE.

Biomicroscopy exam excluded cataracts, and IOLMaster^®^ 700 (Zeiss, Oberkochen, Germany) optical biometry revealed an axial length of 22.71 mm RE and 22.82 mm LE.

Given the clinical diagnosis of IRD, the patient was referred for genetic testing.

### Genetic Testing

The retinal dystrophy mutation analysis report from Manchester Centre for Genomic Medicine [8] identified two heterozygous variants: *RPE65* c.499G>T, p.(Asp167Tyr), likely pathogenic, and *RPE65* c.353G>A, p.(Arg118Lys), variant of uncertain significance. Variant *RPE65* c.499G>T, p.(Asp167Tyr) was also found in the patient’s mother, while the other variant was not detected, indicating that these two variants were on different parental alleles (in *trans*) in our patient. The patient’s father passed away many years ago.

A total of 176 genes were targeted using Agilent Sure Select Custom Design and sequenced on the NextSeq 500 (lllumina, San Diego, CA, USA).

Analytical validation of variants detected by NGS were carried out at the Laboratory of the Department of Medical Genetics, St. Mary’s Hospital, Manchester, in 2017. lndel variants or missense variants that did not meet the internally validated quality criteria were confirmed using Sanger sequencing.

Enrichment was performed with a custom-designed (Retinal dystrophy version 3) Sure Select custom target enrichment kit (Agilent Technologies, Santa Clara, CA, USA) for the NextSeq (lllumina, San Diego, CA, USA) system, following the manufacturer’s protocols. The target enrichment design consisted of the coding region of transcripts, including the immediate splice sites (±5 bases), for 176 genes associated with retinal dystrophy, as detailed below. The samples were sequenced using a NextSeq (Illumina, San Diego, CA, USA), according to the manufacturer’s protocols. Sequence data were aligned to hg19 human genome using BWA-MEM version 0.6.2 (Burrows-Wheeler Alignment Tool, Sanger Institute, Hinxton, Cambridgeshire, UK, University of Hong Kong, Hong Kong, China, Sun Yat-Sen University, Guangdong Province, Guangzhou, Haizhu District, China) and abra version 0.96 (University of North Carolina, Chapel Hill, NC, USA). Variant calling was completed using GenomeAnalysisToolKitLite-version 2.0.39 (GATK, Genome Analysis Tool, Broad Institute, Cambridge, MA, USA) (SNVs and lndels), Pindel version 0.2.4.t (BaseSpace Labs, Illumina, San Diego, CA, USA) (Large lndels) and DeCON version 1.0.1 (Institute of Cancer Research, London, UK, Welcome Trust Centre for Human Genetics, University of Oxford, Oxford, UK) (CNVs). Known polymorphisms were subsequently filtered out of the data obtained using bioinformatic analysis.

In total, >99.9% of the target coding region of the transcripts, as listed below, was covered to a minimum depth of 50×. The DeCON version 1.0.1 program was used to detect copy number variants (CNVs) from targeted NGS data.

Genes tested included:

ABCA4; ABHD12; ACBD5; ADAM9; ADAMTS18; AHl1; AIPL1; ARL2BP; ARL6; BBIP1; BBS1; BBS10; BBS12; BBS2; BBS4; BBS5; BBS7; BBS9; BEST1; C1QTNF5; C2orf71; C20RF86/(WDPCP); C80RF37; C21orf2; CA4; CABP4; CACNA1F; CACNA2D4; CAPN5; CC2D2A; CDH3; CDH23; CDHR1; CEP164; CEP290; CERKL; CHM; CIB2; CLN3; CLRN1; CNGA1; CNGA3; CNGB1; CNGB3; CNNM4; CRB1; CRX; CSPP1; CYP4V2; DFNB31; DHDDS; DTHD1; EFEMP1; ELOVL4; EMC1; EYS; FAM161A; FLVCR1; FSCN2; FZD4; GNAT1; GNAT2; GNPTG; GPR125; GPR179; GPR98; GRM6; GUCA1A; GUCA1B; GUCY2D; HARS; HMX1; IDH3B; IFT140; IMPDH1; IMPG1; IMPG2 7; INPP5E; INVS; IQCB1; ITM2B; KCNJ13; KCNV2; KIAA1549; KIF11; KLHL7; LCA5; LRAT; LRP5; LZTFL1; MERTK; MFRP; MKKS; MKS1; MVK; MYO7A: NDP; NEK2; NMNAT1; NPHP1; NPHP3; NPHP4; NR2E3; NRL; NYX; OAT: OFD1; OTX2; PANK2; PCDH15; PCYT1A; PDE6A; PDE6B; PDE6C; PDE6G; PEX1; PEX2/(PXMP3); PEX7; PHYH; PITPNM3; PLA2G5; PRCD; PROM1; PRPF3; PRPF31; PRPF4; PRPF6; PRPFB: PRPH2, RAB28; RAX2; RBP3; RBP4; RD3; RDH12; RDH5; RGR; RGS9; RHO; RIMS1; RLBP1; ROM1; RP1; RP1L1 (excluding exon 4); RP2; RP9; RPE65; RPGR (excluding ORF15); RPGRIP1; RPGRIP1L; RS1; SAG; SDCCAG8; SEMA4A; SLC24A1; SNRNP200; SPATA7; TEAD1; TIMP3; TMEM237; TOPORS; TRIM32; TRPM1; TSPAN12; TTC8; TUB; TULP1; UNC119; USH1C; USH1G; USH2A; VCAN; VPS13B; WDR19; ZNF423; ZNF513

Intronic mutations tested included: CEP290 c.2991+1665A>G, USH2A c.7595-2144A>G, OFD1 c.935+706A>G, ABCA4 c.5196+1056A>G, ABCA4 c.5196+1137G>A, ABCA4 c.5196+1216C>A, ABCA4 c.4539+2001G>A, ABCA4 c.4539+2028C>T, ABCA4 c.5461-10T>C.

## 3. Discussion

Along with the loss of catalytic activity, lower expression, and rapid degradation of RPE65 [9,10], retinal degeneration might be enhanced by the cytotoxic effect of the mutated RPE65 due to its misfolding, aggregation and mislocalization [11].

There are currently more than 230 variants of *RPE65* known to have been annotated as disease-causing in the HGMD Professional variant database (version 2021.1, accessed on 3 August 2022) [12]. Approximately 60% of the variants are explicitly missense variants, while 40% are truncating variants (i.e., nonsense, frameshift, variants affecting splicing, and gross deletions) [11].

Molecular diagnosis of RP patients is of immense importance due to the existence of targeted treatment with voretigene neparvovec (Luxturna^®^, Novartis, Basel, Switzerland) [6]. The biggest barrier in the interpretation of genetic test results is the presence of VUS [13]. These VUSs populate most genetic testing reports and have increased with the implementation of whole genome sequencing and the growth of test panels [14].

### 3.1. RPE65 c.499G>T, p.(Asp167Tyr)

In the Genome Aggregation Database (gnomAD, accessed on 2 August 2022), a large reference population database that aims to exclude individuals with severe pediatric diseases, this variant was absent from control subjects but was reported at a frequency of 0.000026 in the European (non-Finnish) population [15]. The variant affects a highly conserved amino acid in the carotenoid oxygenase domain of the protein. There is a large physicochemical difference between Asp and Tyr (Grantham score 160, (0–215)), and all in silico tools utilized (Polyphen, SIFT) predict the alteration to be probably damaging and deleterious [15]. *RPE65* c.499G>T, p.(Asp167Tyr) has been previously reported as compound heterozygous with a second disease causing the *RPE65* variant in three patients: with a likely disease-causing frameshift variant in a patient with retinal dystrophy [16]; with a rare, predicted damaging variant, *RPE65* c.938A>G, p.(His313Arg), in a patient with Leber congenital amaurosis [17]; and with a likely disease-causing variant, *RPE65* c.16G>T, p.(Glu6*), in a patient with early-onset, severe retinal dystrophy [18,19]. *RPE65* c.499G>T, p.(Asp167Tyr) was also identified in clinical testing and submitted to ClinVar (Variation ID 98873) [20].

Based on the collective evidence, the variant c.499G>T, p.(Asp167Tyr) is classified as pathogenic/likely pathogenic for *RPE65*-related disorders [20].

### 3.2. RPE65 c.353G>A, p.(Arg118Lys)

This variant has not been identified in normal control populations [16] and in disease-related variation databases such as ClinVar or HGMD (accessed on 3 August 2022) [12,20]. The change affects the final base of the highly conserved genome region of exon 4, and in silico tools (Polyphen) predict the variant to be possibly damaging as it may affect normal splicing of the *RPE65* gene; however, this has to be functionally verified by transcriptional studies [15]. In the GnomAD database (accessed on 2 August 2022), p.(Arg118Lys) is found with a frequency of 0.000003978, and annotated as a missense variation with uncertain clinical significance, predicted to create or disrupt a splice site outside the canonical splice site [15].

Arginine and lysine, two basic amino acids mostly exposed at the protein surface, play important roles in protein stability by forming electrostatic interactions [21]. The Grantham score of 26 classifies this substitution as conservative. However, arginine contains a complex guanidinium group on its side-chain that is ideal for binding negatively charged groups to phosphates, able to form multiple hydrogen bonds and, therefore, cannot be easily substituted by lysine [22]. Moreover, lysine is limited in terms of hydrogen bonds as it only contains a single amino group. A change from arginine to lysine can thus be disastrous [22], and completely eliminate the biological activity of the protein [23].

To the best of our knowledge, *RPE65* c.353G>A, p.(Arg118Lys) is the first described point mutation on this locus, among all other relatively common reported insertional mutations, currently classified as likely benign (ClinVar ID 1557687, 1132389, 1117757) and of uncertain significance (ClinVar 1414138) [15]. Based on the recommendation of the American College of Medical Genetics and Genomics and the Association for Molecular Pathology on standards and guidelines for the interpretation of sequence variants, *RPE65* c.353G>A, p.(Arg118Lys) is: (a) absent in population databases and (b) detected in *trans* with a pathogenic variant, representing moderate evidence of pathogenicity [24]. Moreover, the following determinants, (a) multiple computational evidence supports its deleterious effect, (b) the variant is present in the *RPE65* in which missense variants are a common mechanism of disease and benign missense variants occur at a low rate, and (c) the patient’s phenotype is highly specific for a disease with a single genetic etiology, conform with the supporting pathogenic criteria [24]. In general, combining 2 moderate and ≥2 supporting criteria, the variant *RPE65* c.393T>A, p.(Asn131Lys) could be reclassified as likely pathogenic [24].

This being the case, patients with this specific variant in homozygous or compound heterozygous form would be likely candidates for genetic treatment with voretigene neparvovec [6].

*RPE65*-associated IRD should be suspected in individuals with the following clinical phenotype: onset of symptoms between birth and five years of age, nystagmus or roving eye movements, nyctalopia, decreased BCVA, fundus appearance that tends to be normal in infancy and then ranging from RPE mottling to pigmentary retinopathy with attenuated vessels and optic nerve head pallor, severely diminished or absent fundus autofluorescence, and severely abnormal or barely detectable FFERG [25], all of which are features that correlate with our patient’s phenotype.

Kumaran et al. reported that the 11-year-old female patient who was homozygous for *RPE65* c.353G>A, p.(Arg118Lys) [26]. Based on the severity of the predicted mutational damage of the two null variants causing splice site alteration, the genotype was classified as severe [26]. At 11 years of age, BCVA was 0.6 and 0.64 logMAR at RE and LE, respectively [26]. Our patient, at the age of 40 years, demonstrated a severe disease phenotype with marked macular atrophy (Figure 1 and Figure 2). Even if considered possible, macular atrophy is not the main finding in *RPE65*-related IRD but rather atrophy in the mid and far retinal periphery accompanied by pigment mottling and eventually pigmentary clumps that appear over time [16,27,28]. RP studies observed macular thinning, epiretinal membrane, cystoid macular edema, vitreoretinal interface disorders and choroidal thinning with disease progression [29,30,31,32,33,34]. However, macular atrophy has only been reported in *PROM1* RP and *C8orf37* early-onset retinal dystrophy with cataracts and high myopia [35,36]. No conclusive evidence can be gathered to determine the mechanism and the time of onset and progression of premature macular affectation in the patient presented here. Moreover, the patient is a compound heterozygote, so we cannot clearly define the contribution of each variant to the development of complete macular atrophy. Further reports are needed to elucidate possible correlations between the variants detected and macular atrophy. Regardless, the findings presented here extend the phenotypic spectrum of *RPE65*-associated retinal dystrophies and enrich the knowledge of genotype–phenotype correlations.

## 4. Conclusions

Our patient is a compound heterozygote in the *trans* form and manifests disease with all characteristics typical of the *RPE65*-associated RP [25].

We concluded that this variant contributed to the pathological phenotype, clearly demonstrating its significance, and could be reclassified as likely pathogenic. This being the case, patients with this specific variant in homozygous or compound heterozygous form would be likely candidates for genetic treatment with voretigene neparvovec.

## Figures and Tables

**Figure 1 ijms-23-10513-f001:**
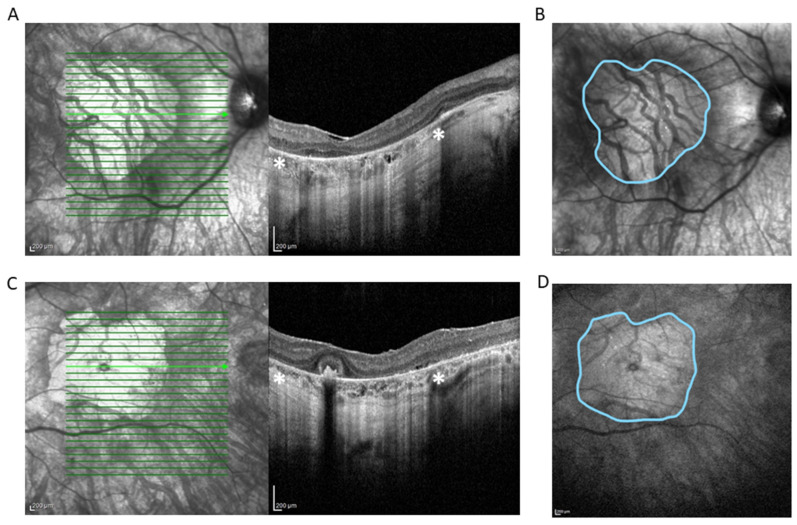
Spectral domain OCT scan (HRA+ OCT Spectralis^®^) depicting complete atrophy of the outer retina, loss of photoreceptors, disruption of retinal pigment epithelium, and underlying choriocapillaris in right eye (RE) (**A**) and left eye (LE) (**C**). * Zone of hypertransmission and sharp signal enhancement below Bruch’s membrane (white asterisks) corresponds to macular atrophy on *en face* near-infrared reflectance image of RE (**B**) and LE (**D**) (outlined in blue). The area of the macular atrophy zone measured with integrated Spectralis^®^ software version 6.8 is 14.75 mm^2^ at RE and 14.45 mm^2^ at LE.

**Figure 2 ijms-23-10513-f002:**
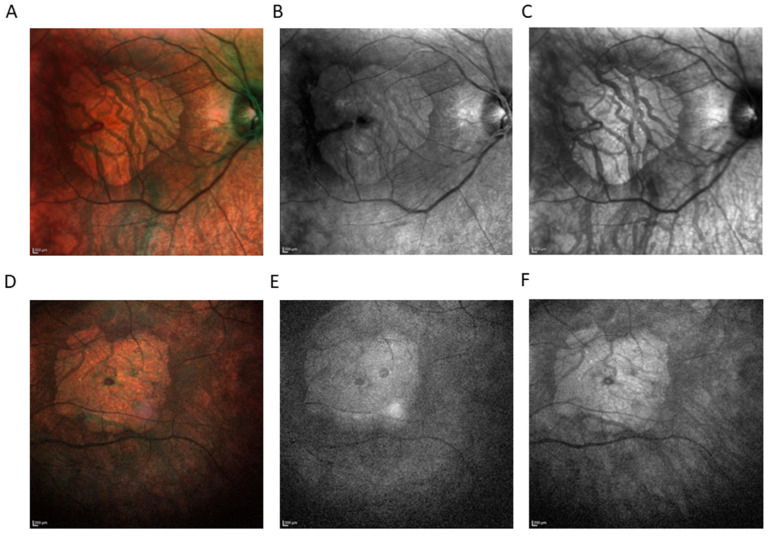
Imaging modality comparison (HRA+ OCT Spectralis^®^) showing area of macular atrophy. Color fundus photography of right eye (RE) (**A**) and left eye (LE) (**D**). Short-wavelength fundus autofluorescence (SW-AF) of RE (**B**), and LE (**E**). Near-infrared reflectance (NIR) of RE (**C**), and LE (**F**). SW-AF was absent, as well as NIR, indicating visual cycle gene mutations.

**Figure 3 ijms-23-10513-f003:**
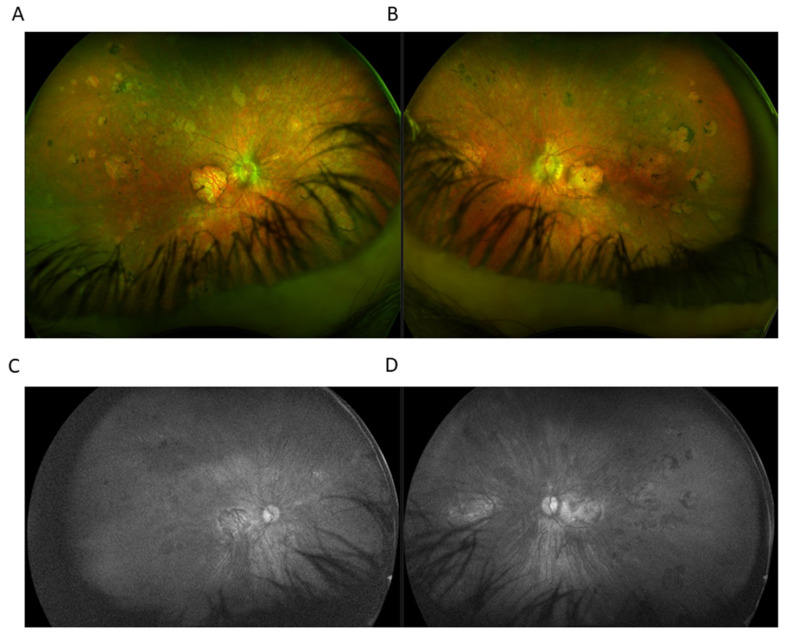
Ultra-widefield (UWF) imaging depicted chorioretinal atrophy in the macular area in right eye (RE) (**A**) and left eye (RE) (**B**). In the mid and far periphery, seals of chorioretinal atrophy with marginal partial pigmentation were present. UWF imaging utilizing green light SW-AF of the RE (**C**) and the LE (**D**) showed the absence of fundus autofluorescence.

## Data Availability

The data presented in this case report are available on request from the corresponding author. The data are not publicly available due to privacy protection.

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
