# Peer review of "RPE65 c.353G>A, p.(Arg118Lys): A Novel Point Mutation Associated with Retinitis Pigmentosa and Macular Atrophy"

_ijms, 2022, doi:10.3390/ijms231810513_

Round 1
Reviewer 1 Report (Previous Reviewer 2)
Bjeloš et al. realized an interesting case report describing the “RPE65 c.353G>A, p.(Arg118Lys): a novel point mutation associated with retinitis pigmentosa and macular atrophy”. I consider the manuscript sufficiently completed, but it requires several revisions needed to improve the completeness of the paper:
· Because the manuscript depicts a point mutation association with retinal phenotype, the results presentation should be improved (e.g. an electropherogram showing found mutation).
· Finally, manuscript requires English revisions and typos correction.
Author Response
Please see the attachment.

Reviewer 2 Report (New Reviewer)
Authors presented an interesting case of a novel RP-related point mutation RPE65 c.353G>A 47 p.(Arg118Lys). They concluded that this variant contributed to the complete macular atrophy.
Although this is a well - written manuscript it's not very clear to me why this mutation is responsible for the macular atrophy. I would recommend that authors clearly clarify this point in the manuscript.
Author Response
Please see the attachment.

This manuscript is a resubmission of an earlier submission. The following is a list of the peer review reports and author responses from that submission.
Round 1
Reviewer 1 Report
Authors describe a case report of early onset IRD in a 40yo man and biallelic RPE65 mutations.
As a frequent concern for gene therapy is the role of VUS in the phenotype, this reviewer agrees on clinical phenotype as a leading proof for VUS pathogenicity. Although well and clearly presented, some observations on the phenotype need to be clarified.
- Although theoretically possible, macular atrophy is not the main finding in RPE65-related IRD: we suggest to include references on phenotype-genotype relationship to support the observation
- Figure 2: B, C and F images are supposed to show SW-AF and NIR of RE respectively and NIR LE: actually, the images resemble more likely only IR images. As correctly stated by authors, absence of SW-AF is one of the leading signs of VC dysfunction in RPE65 so images should clearly demonstrate it
- Authors state that wide field imaging of the fundus has been obtained: I think that showing a widefield picture would be necessary
Reviewer 2 Report
Bjeloš et al. realized an interesting case report describing the “RPE65 c.353G>A, p.(Arg118Lys): a novel point mutation associated with retinitis pigmentosa and macular atrophy”. I consider the manuscript sufficiently completed, but it requires several revisions needed to improve the completeness of the paper:
· Because the manuscript depicts a point mutation association with retinal phenotype, the genetic screening methodology should be improved, as well as results presentation (e.g. an electropherogram showing found mutation).
· Finally, manuscript requires serious English revisions and typos correction.
Round 2
Reviewer 2 Report
Dear authors, even if it is a case report, it is based on genetic results, that must be validated. Such validation are only described and not shown (e.g. electropherograms, in silicon predictions, etc.), and at this stage I consider the manuscript not sufficiently complete to be published.